# Markovian Interference in Experiments

**Vivek Farias**
Sloan School of Management
Massachusetts Institute of Technology
Cambridge, MA 02139
`vivekf@mit.edu`

**Andrew A. Li**
Tepper School of Business
Carnegie Mellon University
Pittsburgh, PA 15213
`aali1@cmu.edu`

**Tianyi Peng**
Department of Aeronautics and Astronautics
Massachusetts Institute of Technology
Cambridge, MA 02139
`tianyi@mit.edu`

**Andrew Zheng**
Operations Research Center
Massachusetts Institute of Technology
Cambridge, MA 02139
`atz@mit.edu`

## Abstract

We consider experiments in dynamical systems where interventions on some experimental units impact other units through a limiting constraint (such as a limited supply of products). Despite outsize practical importance, the best estimators for this 'Markovian' interference problem are largely heuristic in nature, and their bias is not well understood. We formalize the problem of inference in such experiments as one of policy evaluation. Off-policy estimators, while unbiased, apparently incur a large penalty in variance relative to state-of-the-art heuristics. We introduce an on-policy estimator: the Differences-In-Q's (DQ) estimator. We show that the DQ estimator can in general have exponentially smaller variance than off-policy evaluation. At the same time, its bias is second order in the impact of the intervention. This yields a striking bias-variance tradeoff so that the DQ estimator effectively dominates state-of-the-art alternatives. From a theoretical perspective, we introduce three separate novel techniques that are of independent interest in the theory of Reinforcement Learning (RL). Our empirical evaluation includes a set of experiments on a city-scale ride-hailing simulator.

## 1. Introduction

Experimentation is a broadly deployed learning tool in online commerce that is, in principle, simple: apply the treatment in question at random (e.g. an A/B test), and 'naively' infer the treatment effect by differencing the average outcomes under treatment and control. About a decade ago, Blake and Coey [8] pointed out a challenge in such experimentation on Ebay:

*"Consider the example of testing a new search engine ranking algorithm which steers test buyers towards a particular class of items for sale. If test users buy up those items, the supply available to the control users declines."*

The above violation of the so-called Stable Unit Treatment Value Assumption (SUTVA [12]), has been viewed as problematic in the context of online platforms at least as early as Reiley's

36th Conference on Neural Information Processing Systems (NeurIPS 2022).

seminal 'Magic on the Internet' work [34]; Blake and Coey [8] were simply pointing out that the resulting inferential biases were large, which is particularly problematic since treatment effects in this context are typically tiny. The *interference* problem above is germane to experimentation on commerce platforms where interventions on a given experimental unit impact other units since all units effectively share a common inventory of 'demand' or 'supply' depending on context.

Despite what appears to be the ubiquity of such interference, a practical solution is far from settled. The majority of approaches so far fall under the category of *experimental design*, the idea being that a more-careful assignment of treatment will render the bias of the 'naively'-derived inference negligible. This ongoing line of work has produced sophisticated experiment designs which, in the best cases, provably reduce bias under highly specialized models. While this is promising in theory, experimentation on online platforms in particular still largely relies on the simplest designs, i.e. A/B tests. For reasons including cost and organizational frictions, sophisticated experimental designs may not be an ideal lever, and are often infeasible.

**Markovian Interference and Existing Approaches:** We study a generic experimentation problem within a system represented as a Markov Decision Process (MDP), where treatment corresponds to an action which may interfere with state transitions. This form of interference, which we refer to as *Markovian*, naturally subsumes the platform examples above, as recently noted by others either implicitly [41] or explicitly [24, 44]. In that example, a user arrives at each time step, the platform chooses an action (whether to treat the user), and the user's purchase decision alters the system state (inventory levels).

Our goal is to estimate the Average Treatment Effect (ATE), defined as the difference in steady-state reward with and without applying the treatment. In light of the above discussion, we assume that experimentation is done under simple randomization (i.e. A/B testing). Now without design as a lever, there are perhaps two existing families of estimators:

**1. Naive:** We will explicitly define the *Naive* estimator in the next section, but the strategy amounts to simply ignoring the presence of interference. This is by and large what is done in practice. Of course it may suffer from high bias (we show this formally in Example 1), but it serves as more than just a strawman. In particular, bias is only one side of the estimation coin, and with respect to the other side, namely variance, the Naive estimator is effectively the best possible.

**2. Off-Policy Evaluation (OPE):** Another approach comes from viewing our problem as one of policy evaluation in reinforcement learning (RL). Succinctly, it can be viewed as estimating the average reward of two different policies (no treatment, or treatment) given observations from some *third* policy (simple randomization). This immediately suggests framing the problem as one of *Off-Policy Evaluation*, and borrowing one of many existing *unbiased* estimators, e.g. [50, 49, 33, 22, 27, 28]. This tack appears to be promising, e.g. [44], but we observe that the resulting variance is necessarily large (Theorem 3).

**Our Contributions:** Against the above backdrop, we propose a novel *on*-policy treatment-effect estimator, which we dub the 'Differences-In-Q's (DQ)' estimator, for experiments with Markovian interference. In a nutshell, we characterize our contribution as follows:

*The DQ estimator has provably negligible bias relative to the treatment effect. Its variance can, in general, be exponentially smaller than that of an efficent off-policy estimator. In both stylized and large-scale real-world models, it dominates state-of-the-art alternatives.*

We next describe these relative merits in greater detail:

**1. Second-order Bias:** We show (Theorem 1) that when the impact of an intervention on transition probabilities is $O(\delta)$, the bias of the DQ estimator is $O(\delta^2)$. The DQ estimator thus leverages the one piece of structure we have relative to generic off-policy evaluation: treatment effects are typically small.

**2. Variance:** We show (Theorem 2) that the DQ estimator is asymptotically normal, and provide a non-trivial, explicit characterization of its variance. By comparison, we show (Theorem 4) that this variance can, in general, be exponentially (in the size of the state space) smaller than the variance of *any* unbiased off-policy estimator.

Summarizing the above points, we are the first (to our knowledge) to explicitly characterize the favorable bias-variance trade-off in using *on-policy* estimation to tackle off-policy evaluation. This new lens has broader implications for OPE and policy optimization in RL.

**3. Practical Performance:** We conduct experiments in both a caricatured one-dimensional environment proposed by others [24], as well as a city-scale simulator of a ride-sharing platform. We show that in both settings the DQ estimator has MSE that is substantially lower than (a) naive and off-policy estimators, and even (b) estimators given access to incumbent state-of-the-art experimental *designs*.

**Related Literature:** The largest portion of work in interference is in *experimental design*, with the design levers ranging from stopping times in A/B tests [30, 23, 57, 25], to any form of more-sophisticated 'clustering' of units [11, 16, 19, 14, 37, 53, 55, 15], to clustering specifically when interference is represented by a network [35, 54, 42, 2, 7, 39, 60], to the proportion of units treated [21, 48, 4], to the timing of treatment [45, 9, 17], and beyond [3, 29, 51, 35, 10, 6, 20, 42]. As alluded to earlier, these sophisticated designs can be powerful, but cost, user experience, and other implementation concerns restrict their application in practice [31, 32].

We view this paper as orthogonal to this literature, but will eventually compare against a recent state-of-the-art design, so-called *two-sided randomization* [24, 5], that is specific to the context of two-sided marketplaces (e.g. the one we simulate).

As stated earlier, the problem we study is one of *off-policy evaluation (OPE)* [40, 46]. The fundamental challenge in OPE is high variance, which can be attributed to the nature of the algorithmic tools used, e.g. sampling procedures [50, 49, 33]. Recent work on 'doubly-robust' estimators [22, 27, 28] has improved on variance (incidentally, our estimator is loosely tied to these, as we discuss in Section 6), but again we will show, via a formal lower bound, that unbiased estimators as a whole have prohibitively large variance. Finally, our motivation is close in spirit to a recent paper [44], which applies OPE directly in Markovian interference settings; we make a direct experimental comparison in Section 5.

In the policy optimization literature, 'trust-region' methods [43] and conservative policy iteration [26] use a related on-policy estimation approach to bound policy improvement. However, the explicit application of on-policy estimation in the context of OPE, and in particular the striking bias-variance tradeoff this enables, are novel to this paper.

## 2. Model

This section formalizes the inference problem that we tackle, casting it in the language of MDPs. Vis-à-vis the existing literature, this lens allows us to reason about the problem using a large, well-established toolkit, and makes obvious the fact that OPE provides unbiased estimation of the ATE. We then present what we call the 'Naive' estimator (alluded to in the introduction). This is the lowest-variance estimator one can hope for in this setting, but it can have significant bias, as we will see.

We begin by defining an MDP with state space $\mathcal{S}$. We denote by $s_t \in \mathcal{S}$ the state of the MDP at time $t \in \mathbb{N}$. Every state is associated with a set of available actions $\mathcal{A}$ which govern the transition probabilities between states via the (unknown) function $p : \mathcal{S} \times \mathcal{A} \times \mathcal{S} \rightarrow [0, 1]$. We assume that $\mathcal{A} = \{0, 1\}$ irrespective of state; for descriptive purposes, we will associate the '1' action with the use of a prospective intervention, so that '0' is associated with not employing the intervention. We denote by $r(s, a)$ the reward earned in state $s$ having employed action $a$. A policy $\pi : \mathcal{S} \rightarrow \mathcal{A}$ maps states to random actions. We define the average reward $\lambda^\pi$, under any (ergodic, unichain) policy $\pi$, according to:

$$\lambda^\pi = \lim_{T \to \infty} \frac{1}{T} \sum_{t=1}^{T} r(s_t, \pi(s_t)).$$

There are three policies we define explicitly:

**The Incumbent Policy $\pi_0$:** This policy never uses the intervention, so that $\pi_0(s) = 0$ for all $s$. This is 'business as usual'. Denote the associated transition matrix as $P_0$ (i.e. the entries of $P_0$ are exactly $p(\cdot, 0, \cdot)$)

**The Intervention Policy $\pi_1$:** This policy always uses the intervention, so that $\pi_1(s) = 1$ for all $s$. This reflects the system, should the intervention under consideration be 'rolled out'. Denote the associated transition matrix as $P_1$.

**The Experimentation Policy $\pi_p$:** This policy corresponds to the experiment design. Simple randomization would select $\pi(s) = 1$ with some fixed probability $p$, say $1/2$, independently at every period. This corresponds to the sort of search engine experiment alluded to in the introduction. The transition matrix associated with this design is then $P_{1/2} = \frac{1}{2}P_0 + \frac{1}{2}P_1$.

**The Inference Problem:** We are given a single sequence of $T$ states, actions, and rewards, observed under the experimentation policy $\pi_p$ (recall that cost and constraints [31, 32] prohibit us from running $\pi_0$ or $\pi_1$ separately until convergence). The data we have is the sequence $\{(s_t, a_t, r(s_t, a_t)) : t = 1, \ldots, T\}$, wherein $a_t \triangleq \pi_p(s_t)$. We must estimate the average treatment effect (ATE):

$$\text{ATE} \triangleq \lambda^{\pi_1} - \lambda^{\pi_0}.$$

## 2.1. The Naive Estimator and Bias

A natural approach to estimating the ATE is to use simple randomization (i.e. $P_{1/2}$) and the following *Naive* estimator:

$$(1) \qquad \hat{\text{ATE}}_N = \frac{1}{|T_1|} \sum_{t \in T_1} r(s_t, a_t) - \frac{1}{|T_0|} \sum_{t \in T_0} r(s_t, a_t),$$

where $T_1 = \{t : a_t = 1\}$ and $T_0 = \{t : a_t = 0\}$. In the context of the search engine experiment, this corresponds to simply averaging some metric of interest (say, conversion) among the test users ($T_1$) and control users ($T_0$). What goes wrong is simply that the two empirical averages above, that seek to estimate $\lambda^{\pi_1}$ and $\lambda^{\pi_0}$ respectively, employ the wrong measure over states. This is sufficient to introduce bias that is on the order of the treatment effect being estimated:

**Example 1.** *Consider an MDP on two states, $\mathcal{S} = \{\mathbf{0}, \mathbf{1}\}$. We collect a reward of $0$ in state $\mathbf{0}$ irrespective of the action taken in that state ($r(\mathbf{0}, 0) = r(\mathbf{0}, 1) = 0$), and a reward of $1$ in state $\mathbf{1}$, again, irrespective of action ($r(\mathbf{1}, 0) = r(\mathbf{1}, 1) = 1$). On the other hand, transitions are impacted by our choice of action. Specifically, let $p(\mathbf{0}, 0, \mathbf{0}) = p(\mathbf{0}, 0, \mathbf{1}) = p(\mathbf{1}, 0, \mathbf{1}) = p(\mathbf{1}, 0, \mathbf{0}) = 1/2$. We maintain $p(\mathbf{0}, 1, \mathbf{1}) = p(\mathbf{0}, 1, \mathbf{0}) = 1/2$ so that the intervention has no effect at state $\mathbf{0}$. On the other hand, we let $p(\mathbf{1}, 1, \mathbf{1}) = 1/2 + \delta$, so that $p(\mathbf{1}, 1, \mathbf{0}) = 1/2 - \delta$, for some $\delta > 0$. In words, the intervention tends to discourage a transition to $\mathbf{0}$ from state $\mathbf{1}$.*

In the above example, it is easy to calculate that $\text{ATE} = (1/2)\delta/(1 - \delta)$, reflecting the shift in the stationary distribution favoring state $\mathbf{1}$, induced under the intervention. On the other hand, we can calculate that $\lim_T \hat{\text{ATE}}_N = 0$, so that the bias induced by the 'experimentation' policy relative to the stationary distributions under the incumbent and intervention policies respectively, is comparable to the size of the treatment effect.

## 3. The Differences-In-Q's Estimator

We are now prepared to introduce our estimator for inference in the presence of Markovian interference. Before defining our estimator, which we will see is only slightly more complicated than the Naive estimator, we recall a few useful objects associated with MDPs. First, for a fixed policy $\pi$, define the Bellman operator $T_\pi : \mathbb{R}^{|\mathcal{S}|} \times \mathbb{R} \to \mathbb{R}^{|\mathcal{S}|}$ according to

$$T_\pi(V, \lambda) = r_\pi - \lambda \mathbf{1} + P_\pi V,$$

where $r_\pi : \mathcal{S} \to \mathbb{R}$ is defined according to $r_\pi(s) = \mathsf{E}\left[r(s, \pi(s))\right]$. The average cost of policy $\pi$, denoted $\lambda^\pi$, and the bias function corresponding to $\pi$, denoted $V_\pi$, are then a solution to the fixed point equation $T_\pi(V, \lambda) = V$. Finally, the $Q$-function associated with $\pi$, denoted $Q_\pi : \mathcal{S} \times \mathcal{A} \to \mathbb{R}$, is defined according to

$$(2) \qquad Q_\pi(s, a) = r(s, a) - \lambda^\pi + \mathsf{E}\left[V_\pi(s_1)|s_0 = s, a_0 = a\right].$$

## 3.1. An Idealized First Step

In motivating our estimator, let us begin with the following idealization of the Naive estimator, where we denote by $\rho_{1/2}$ the steady state distribution under the randomization policy $\pi_{1/2}$:

$$\mathsf{E}_{\rho_{1/2}}\left[\hat{\mathrm{ATE}}_N\right] = \sum_s \rho_{1/2}(s)\left[r(s,1) - r(s,0)\right].$$

It is not hard to see that in the context of Example 1, we continue to have $\mathsf{E}_{\rho_{1/2}}[\hat{\mathrm{ATE}}_N] = 0$, so that this idealization of the Naive estimator continues to have bias on the order of the treatment effect. Consider then, the following alternative:

$$(3) \qquad \mathsf{E}_{\rho_{1/2}}\left[\hat{\mathrm{ATE}}_D\right] = \sum_s \rho_{1/2}(s)\left[Q_{\pi_{1/2}}(s,1) - Q_{\pi_{1/2}}(s,0)\right],$$

where the term $\mathsf{E}_{\rho_{1/2}}[\hat{\mathrm{ATE}}_D]$ can for now just be thought of as an idealized constant ($\hat{\mathrm{ATE}}_D$ is defined soon in (4)). Compared to $\mathsf{E}_{\rho_{1/2}}[\hat{\mathrm{ATE}}_N]$, we see that $\mathsf{E}_{\rho_{1/2}}[\hat{\mathrm{ATE}}_D]$ takes a remarkably similar form, except that as opposed to an average over differences in rewards, we compute an average of differences in $Q$-function values. The idea is that doing so will hopefully compensate for the shift in distribution induced by $\pi_{1/2}$. We return to our example to check:

**Example 1** (Continued). *Continuing with our example, we can explicitly calculate $Q_{\pi_{1/2}}(\cdot,\cdot)$, the average reward $\lambda^{\pi_{1/2}}$, and the stationary distribution $\rho_{1/2}$ (see Appendix[1]). Doing so allows us to calculate that*

$$\mathsf{E}_{\rho_{1/2}}\left[\hat{\mathrm{ATE}}_D\right] = \frac{1}{2}\left(\frac{\delta}{(1-\delta/2)^2}\right).$$

*That is, $|\mathrm{ATE} - \mathsf{E}_{\rho_{1/2}}[\hat{\mathrm{ATE}}_D]| = O(\delta^2)$, so that the bias of this idealized estimator is second-order (i.e. negligible) relative to the ATE.*

Is the dramatic mitigation of bias we see in Example 1 generic? If the experimentation policy mixes fast, our first set of results essentially answers this question in the affirmative. In particular, we make the following mixing time assumption:

**Assumption 1** (Mixing time). *There exist constants $C$ and $\lambda$ such that for all $s \in \mathcal{S}$ and for any integer $k \geq 0$,*

$$d_{\mathrm{TV}}(P_{1/2}^k(s,\cdot), \rho_{1/2}) \leq C\lambda^k,$$

*where $d_{\mathrm{TV}}(\cdot,\cdot)$ denotes total variation distance.*

We then have that the second order bias we saw in Example 1 is, in fact, generic:

**Theorem 1** (Bias of DQ). *Assume that for any state $s \in \mathcal{S}$, $d_{\mathrm{TV}}(p(s,1,\cdot), p(s,0,\cdot)) \leq \delta$. Then,*

$$\left|\mathrm{ATE} - \mathsf{E}_{\rho_{1/2}}\left[\hat{\mathrm{ATE}}_D\right]\right| \leq C'\left(\frac{1}{1-\lambda}\right)^2 r_{\max} \cdot \delta^2$$

*where $r_{\max} := \max_{s,a}|r(s,a)|$ and $C'$ is a constant depending (polynomially) on $\log(C)$.*

## 3.2. The Differences-In-Q's Estimator

Motivated by the development in the previous subsection, the *Differences-In-Q's (DQ)* estimator we propose to use is simply

$$(4) \qquad \hat{\mathrm{ATE}}_D = \frac{1}{|T_1|}\sum_{t \in T_1}\hat{Q}_{\pi_{1/2}}(s_t, a_t) - \frac{1}{|T_0|}\sum_{t \in T_0}\hat{Q}_{\pi_{1/2}}(s_t, a_t),$$

---

[1]Appendices can be found at https://arxiv.org/abs/2206.02371.

where we take an empirical average over the state trajectory produced under the randomization policy, and $\hat{Q}_{\pi_{1/2}}$ is an estimator of the $Q$-function. For concreteness, we obtain $\hat{Q}_{\pi_{1/2}}$ by solving

$$(5) \qquad \min_{\hat{V}, \hat{\lambda}} \sum_{s \in \mathcal{S}} \left( \sum_{t, s_t = s} r(s_t, a_t) - \hat{\lambda} + \hat{V}(s_{t+1}) - \hat{V}(s_t) \right)^2.$$

Our main result characterizes the variance and asymptotic normality of $\hat{\text{ATE}}_D$:

**Theorem 2** (Variance and Asymptotic Normality of DQ). *The DQ estimator is asymptotically normal so that*

$$\sqrt{T} \left( \hat{\text{ATE}}_D - \mathsf{E}_{\rho_{1/2}} \left[ \hat{\text{ATE}}_D \right] \right) \xrightarrow{d} \mathcal{N}(0, \sigma_D^2),$$

*with standard deviation*

$$\sigma_D \leq C' \left( \frac{1}{1 - \lambda} \right)^{5/2} \log \left( \frac{1}{\min_{s \in S} \rho_{1/2}(s)} \right) r_{\max}.$$

*where $C'$ is a constant depending (polynomially) on $\log(C)$.*

**One Extreme of the Bias-Variance Tradeoff:** We may heuristically think of the Naive estimator as representing one extreme of the bias-variance tradeoff among reasonable estimators. For the sake of comparison, by the Markov Chain CLT, the Naive estimator is also asymptotically normal with standard deviation $\Theta(r_{\max}/(1 - \lambda)^{1/2})$. This rate is efficient for the estimation of the mean of a Markov chain [18]. On the other hand, while the Naive estimator is effectively useless for the problem at hand given its bias is in general $\Theta(\delta)$, that of the DQ estimator is $O(\delta^2)$.

## 3.3. A series expansion for the ATE

Our key technical contribution, as well as the motivation for the DQ estimator, is a novel Taylor series representation of the ATE which describes the *off-policy* average reward as a sum of terms that can be estimated *on-policy*. The Naive estimator emerges as the zeroth-order truncation of the series; and the idealized DQ estimator is the natural first-order correction. The proof of Theorem 1 proceeds by bounding the remainder. Here we sketch the deriviation of this series; see the Appendix for the full proof.

We first define few pieces of useful notation. Let $\rho_0 \in \mathbb{R}^{|\mathcal{S}|}, \rho_{1/2} \in \mathbb{R}^{|\mathcal{S}|}, \rho_1 \in \mathbb{R}^{|\mathcal{S}|}$ be the vectors of the stationary distributions of $P_0, P_{1/2}, P_1$ accordingly. Let $r_0 \in \mathbb{R}^{|\mathcal{S}|}, r_{1/2} \in \mathbb{R}^{|\mathcal{S}|}, r_1 \in \mathbb{R}^{|\mathcal{S}|}$ be the reward vectors associated with policies $\pi_0, \pi_{1/2}, \pi_1$, i.e., $r_a(s) = r(s, a)$ and $r_{1/2} = \frac{1}{2} r_0 + \frac{1}{2} r_1$. The proof relies on a classic perturbation result, which allows us to quantify exactly the error induced by distribution shift:

**Lemma 1** (Stationary Distribution Perturbation [36]). *Let $P, P' \in \mathbb{R}^{|\mathcal{S}| \times |\mathcal{S}|}$ be kernels of aperiodic and irreducible Markov Chains, with stationary distributions $\rho, \rho' \in \mathbb{R}^{|\mathcal{S}|}$. Then,*

$$\rho'^{\top} = \rho^{\top} + \rho'^{\top} (P' - P)(I - P)^{\#}$$

*where $(I - P)^{\#} = (I - P + \mathbf{1}\rho^{\top})^{-1} - \mathbf{1}\rho^{\top}$ is the group inverse of $I - P$.*

We will first apply Lemma 1 to express $\lambda_1 = \rho_1^{\top} r_1$ in terms of $\rho_{1/2}$:

$$\rho_1^{\top} r_1 = \rho_{1/2}^{\top} r_1 + \rho_1^{\top} (P_1 - P_{1/2})(I - P_{1/2})^{\#} r_1$$

We can now apply Lemma 1 again to $\rho_1$ in the RHS; applying this $K$ times iteratively yields the expansion:

$$\rho_1^{\top} r_1 = \sum_{k=0}^{K} \rho_{1/2}^{\top} \left[ (P_1 - P_{1/2})(I - P_{1/2})^{\#} \right]^k r_1 + \rho_1^{\top} \left[ (P_1 - P_{1/2})(I - P_{1/2})^{\#} \right]^{K+1} r_1$$

This expansion expresses the *off-policy* average reward as a sum of *on-policy* quantities (i.e., expectations under $\rho_{1/2}$), plus a remainder term that can be bounded as:

$$\left| \rho_1^\top \left[ (P_1 - P_{1/2})(I - P_{1/2})^\# \right]^{K+1} r_1 \right| \leq C' \left( \frac{\delta}{1 - \lambda} \right)^{K+1} r_{\max}$$

This has several immediate implications. First, the Naive estimator converges to the difference between the zeroth terms of the expansion (i.e., $K = 0$) for $\rho_1^\top r_1$ and $\rho_1^\top r_0$; the resulting bias is then $O(\delta)$. The DQ estimator, in contrast, converges to the difference between the first-order partial sums of the expansion (i.e., $K = 1$) for $\rho_1^\top r_1$ and $\rho_1^\top r_0$; the $O(\delta^2)$ bias in Theorem 1 then follows immediately. This strategy can be iterated to generalize the DQ estimator to arbitrarily high-order bias corrections.

## 4. The Price of Being Unbiased

Thus far, we have seen that the DQ estimator provides a dramatic mitigation in bias (Theorem 1) at a relatively modest price in variance (Theorem 2). This suggests another question: could we hope to construct an *unbiased* estimator that has low variance (i.e. comparable to either the Naive or DQ estimators). We will see that the short answer is: no.

### 4.1. The Variance of an Optimal Unbiased Estimator

As noted earlier, a plethora of Off-policy evaluation (OPE) algorithms might be used to provide an unbiased estimate of the ATE. Rather than consider a particular OPE algorithm, here we produce a lower bound on the variance of *any* unbiased OPE algorithm. While such a bound is obviously of independent interest (since OPE is a far more general problem than what we seek to accomplish in this paper), we will primarily be interested in comparing this lower bound to the variance of the DQ estimator from Theorem 2.

**Theorem 3** (Variance Lower Bound for Unbiased Estimators). *Assume we are given a dataset* $\{(s_t, a_t, r(s_t, a_t)) : t = 0, \ldots, T\}$ *generated under the experimentation policy* $\pi_{1/2}$, *with* $s_0$ *distributed according to* $\rho_{1/2}$. *Then for any unbiased estimator* $\hat{\tau}$ *of* ATE, *we have that*

$$T \cdot \mathrm{Var}(\hat{\tau}) \geq 2 \sum_s \frac{\rho_1(s)^2}{\rho_{1/2}(s)} \sum_{s'} p(s, 1, s')(V_{\pi_1}(s') - V_{\pi_1}(s) + r(s, 1) - \lambda^{\pi_1})^2$$

$$+ 2 \sum_s \frac{\rho_0(s)^2}{\rho_{1/2}(s)} \sum_{s'} p(s, 0, s')(V_{\pi_0}(s') - V_{\pi_0}(s) + r(s, 0) - \lambda^{\pi_0})^2 \triangleq \sigma_{\mathrm{off}}^2.$$

It is worth remarking that this lower bound is tight: in the appendix we show that an LSTD(0)-type OPE algorithm achieves this lower bound. While this is of independent interest vis-à-vis average cost OPE, we turn next to our ostensible goal here – evaluating the 'price' of unbiasedness. We can do so simply by comparing the variance of the DQ estimator with the lower bound above. In fact, we are able to exhibit a class of one-dimensional Markov chains (in essence the same model proposed by [24] as a caricature of the dynamic interference problem) for which we have:

**Theorem 4** (Price of Unbiasedness). *For any* $0 < \delta \leq \frac{1}{5}$, *there exists a class of MDPs parameterized by* $n \in \mathbb{N}$, *where* $n$ *is the number of states, such that*

$$\frac{\sigma_D}{\sigma_{\mathrm{off}}} = O\left( \frac{n^8}{c^n} \right),$$

*for some constant* $c > 1$. *Furthermore,* $|(\mathrm{ATE} - \mathsf{E}[\hat{\mathrm{ATE}}_D])/\mathrm{ATE}| \leq \delta$.

**Another Extreme of the Bias-Variance Tradeoff:** Theorems 2, 3, and 4 together reveal the opposite extreme of the bias-variance tradeoff. Specifically, if we insisted on an unbiased estimator for our problem (of which there are many, thanks to our framing of the problem as one of OPE), we would pay a large price in terms of variance. In particular Theorem 4

illustrates that this price can grow exponentially in the size of the state space. This jibes with our empirical evaluation in both caricatured and large-scale MDPs in Section 5.

Taken together our results reveal that the DQ estimator accomplishes a striking bias-variance tradeoff: it has substantially smaller variance than any unbiased estimator (in fact, comparable to the Naive estimator), all while ensuring bias that is second order in the impact of the intervention.

# 5. Experiments

This section will empirically investigate the DQ estimator and a number of alternatives in two settings: a simple one-dimensional toy model proposed by [24], and more realistically, a city-scale simulator of a ride-hailing platform similar to what large ride-hailing operators use in production. The alternatives we consider include: 1) the Naive estimator; 2) TSRI-1 and TSRI-2, the "two-sided randomization" (TSR) designs/estimators from [24]; 3) a variety of OPE estimators. For the OPE estimators, we note that off-policy average reward estimation has only recently been addressed in [56, 59], and we implement their specific estimators which we simply denote as TD and GTD respectively. We also implement an extension to an LSTD type estimator proposed in [44].

## 5.1. A toy example

We first study all of our estimators in a simple setting that does not call for any sort of value function approximation. Our goal is to understand the relative merits of these estimators in terms of their bias and variance. To this end, we adopt precisely the toy MDP studied by [24]; a stylized model of a rental marketplace. This MDP is essentially a 1-D Markov chain on $N = 5000$ states parameterized by a 'customer arrival' rate $\lambda$ and a 'rental duration' rate $\mu$. At a given state $n$ (so that $n$ units of inventory are in the system), the probability that an arriving customer rents a unit is impacted by the intervention. As such if the intervention increases the probability of a customer renting, this reduces the inventory availability for customers that arrive later. Our MDP setup exactly replicates that of [24], with $N = 5000, \lambda = 1, \mu = 1$; see the appendix for further details. We run all estimators over

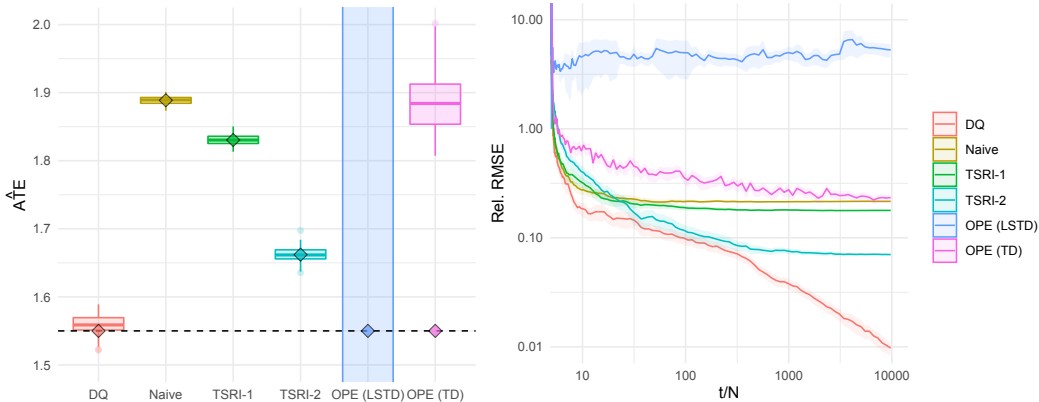

**Figure 1:** Toy-example from [24]. *Left*: Estimated ATE at time $t/N = 10^4$ across 100 trajectories. Dashed line indicates actual ATE. Diamonds indicate the asymptotic mean for each estimator. DQ shows compelling bias-variance tradeoff for this experimental budget. *Right*: Relative RMSE vs. Time; DQ dominates the alternatives at all timescales.

100 separate trajectories of length $t = 10^4 N$ of the above MDP initialized in its stationary distribution. Figure 1 summarizes the results of this experiment. Beginning with the left panel, which reports estimated quantities at $t = 10^4 N$, we immediately see:
**TSR improves on Naive:** The actual ATE in the experiment is 1.5%. Whereas it has the lowest variance of the estimators here, the Naive estimator has among the highest bias. The

two TSR estimators reduce this bias substantially at a modest increase in variance. It is worth noting, as a sanity check, that these results precisely recreate those reported in [24]. **OPE estimators are high variance:** The OPE estimators have the highest variance of those considered here. The TD estimator has the lower variance but this is simply because it is implicitly regularized. Run long enough, both estimators will recover the treatment effect. **DQ shows a compelling bias-variance tradeoff:** In contrast, the DQ estimator has the lowest bias at $t = 10^4 N$ and its variance is comparable to the TSR estimators (It is worth noting that run long enough, the DQ estimator had a bias of $\sim -5 \times 10^{-7}$). **Conclusions hold across experimental budgets:** Turning our attention briefly to the right chart in Figure 1, we show the relative RMSE (i.e. RMSE normalized by the treatment effect) of the various estimators considered here *across all experimental budgets $t$*. RMSE effectively scalarizes bias and variance and we see that on this scalarization the DQ estimator dominates the other estimators considered here over all choice of $t$.

We note that specialized designs such as TSR can still be valuable in specific settings: when $\lambda \gg \mu$, for example, TSR is nearly unbiased (as shown in [24]), and can outperform DQ; see the appendix for such a study.

## 5.2. A Large-Scale Ridesharing Simulator

We next turn our attention to a city-scale ridesharing simulator similar to those used in production at large ride-hailing services. We will consider the problem of experimenting with changes to *dispatching* rules. Experimenting with these changes naturally creates Markovian interference by impacting the downstream supply/ positioning of drivers. Relative to the earlier toy example, the corresponding MDP here has an intractably large state-space, necessitating value function approximation for the DQ and OPE estimators.

**The simulator:** Ridesharing admits a natural MDP; see e.g. [41]. The state at the time of a request corresponds to that of all drivers at that time: position, assigned routes, riders, and the pickup/dropoff location of the request. Actions correspond to driver assignments and pricing decisions. The reward for a request is the price paid by the rider, less cost incurred to service the request. Our simulator models Manhattan. Riders and drivers are generated according to real world data, based on [1]; this yields $\sim 300k$ requests and $\sim 7k$ unique drivers per real day. An arriving request is served a menu of options generated by a price engine. The rider chooses an option based on a choice model calibrated on taxi prices (for the outside option) and implied delay disutility from typical match rates. A dispatch engine assigns a driver to the rider; the engine chooses the driver who can serve the rider at minimal marginal cost, subject to the product's constraints. Finally drivers proceed along their assigned routes until the next request is received. The simulator implements pooling. Users can switch out demand and supply generation, pricing and dispatch algorithms, driver repositioning, and the choice model via a simple API. Other simulators exist in the literature [41, 58], but lack either an open-source implementation, or implement a subset of the functionality here.

**The experiment:** We experiment with dispatch policies. Specifically, we consider assigning a request to an idle driver or a 'pool' driver, i.e. a driver who already has riders in their car. A dispatch algorithm might prefer the former, but only if the cost of the resulting trip is at most $\alpha\%$ higher than the cost of assigning to a pool driver. We consider three experiments, each of which changes $\alpha$ from a baseline of 0 to one of three distinct values: $30\%, 50\%$ or $70\%$, with ATEs of 0.5%, -0.9%, and -4.6% respectively. As we noted earlier, we would expect significant interference in this experiment (or indeed any experiment that experiments with pricing or dispatch) since an intervention changes the availability / position of drivers for subsequent requests.

Figure 2 summarizes the results of the above experiments, wherein each estimator was run over 50 independent simulator trajectories, each over $3 \times 10^5$ requests. The DQ and OPE estimators shared a common linear approximation architecture with basis functions that count the number of drivers at every occupancy level. We note that this approximation introduces its own bias which is not addressed by our theory. We immediately see: **Strong Impact of Interference:** As we might expect, interference has a significant impact here as witnessed by the large bias in the Naive estimator. **Incumbent estimators do not improve on Naive:** None of the incumbent estimators

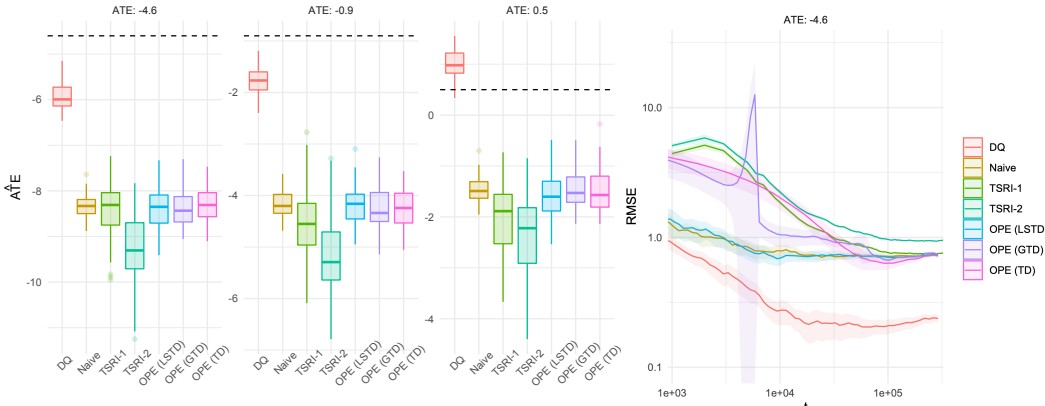

**Figure 2:** Ridesharing model *Left:* $\hat{\text{ATE}}$ at $t = 3 \times 10^5$ over 50 trajectories. Dashed line indicates actual ATE. DQ has lowest bias, and is only estimator to estimate correct sign of the treatment at all effect sizes. *Right:* RMSE vs. Time; DQ dominates at all time scales.

improve on Naive in this hard problem. This is also the case for the TSR designs, which in this large scale setting surprisingly appear to have significant variance. The OPE estimators have lower variance due to the regularization caused by value function approximation.

**DQ works:** In all three experiments, the bias in DQ (although in a relative sense higher than in the toy model) is *substantially* smaller than the alternatives, and also smaller than the ATE. This is evident in the left panel in Figure 2. Notice that in the rightmost experiment (ATE = 0.5), DQ is the only estimator to learn that the ATE is positive. Like in the toy model, the right panel shows that these results are robust over experimentation budgets.

# 6. Discussion: refining the bias-variance tradeoff

To summarize, we have shown that the DQ estimator achieves a surprising bias-variance tradeoff by applying on-policy estimation to the Markovian interference problem, and more generally to OPE. Here we draw further connections between the Naive, DQ, and OPE estimators, and suggest how to interpolate between these estimators to realize other points along the bias-variance curve.

**State-space aggregation and the Naive estimator:** Consider approximate estimation of the value function via state aggregation, for example in cases where the state space is massive. At one extreme, the DQ estimator corresponds to performing no aggregation; at the other, using a single aggregate state reproduces the Naive estimator exactly. Controlling the level of aggregation (or more generally, the complexity of the value function approximation) interpolates between the DQ and Naive estimators.

**Regularization and the OPE meta-estimator:** A large family of OPE estimators are formulated explicitly based on the following identity on the ATE:

$$(6) \qquad \text{ATE} = \mathsf{E}_{\rho_{1/2}} \left[ \frac{(1/2)(\rho_1(s) + \rho_0(s))}{\rho_{1/2}(s)} (Q_{\pi_{1/2}}(s, 1) - Q_{\pi_{1/2}}(s, 0)) \right]$$

Doubly robust estimators (see e.g. [27, 52]), for example, plug in estimates of the likelihood ratio and value functions to (5), as do the closely related primal-dual estimators (see [13, 47]). However our theory (Theorem 3) demonstrates that the cost of this unbiased estimation is prohibitively high.

Notice, however, that simply setting the likelihood ratio to one immediately recovers the DQ estimator – and its favorable guarantees. By regularizing the likelihood ratio estimates in OPE methods towards one, then, one can interpolate along the bias-variance curve between DQ and unbiased OPE. Recent regularized primal-dual OPE algorithms (e.g. [38]) have in fact realized empirical performance gains from such regularization.

**Higher-order Differences-in-Qs:** As alluded to in Section 3.3, the series expansion that motivates the DQ estimator can be extended to obtain arbitrarily high-order bias corrections. The resulting estimators admit natural interpretations as *Differences-in-Qs-of-Differences-in-Qs*. This series of bias corrections represents a natural interpolation along the full bias-variance curve, from Naive estimation to unbiased OPE. Analyzing the bias-variance tradeoff along this curve is an exciting direction for future work.

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
