# OpenReview forum: "Markovian Interference in Experiments"
_NeurIPS.cc/2022/Conference — NeurIPS 2022 Accept_

### Official Review · Reviewer_frjR · 2022-07-11

**Rating:** 6
**Confidence:** 3
**Soundness:** 3 good
**Presentation:** 3 good
**Contribution:** 3 good

**Summary:**

This paper develops a novel biased estimator for the average treatment effect with an MDP model. Here, the average treatment effect is the difference between the reward obtained with and without treatment. This estimator is the difference between the empirical average of Q-values corresponding to the cases with and without applying treatment, which is very similar to a naive estimator that computes the difference of rewards instead of Q-values. However, the authors show that in some cases, this new estimator has a negligible bias, which is significantly better than the naive estimator. Moreover, with some mild conditions, they show that this new estimator is asymptotically unbiased. The variance of the limit random variable of this new estimator enjoys a smaller variance than all unbiased estimators. Finally, the authors elaborate their results with numerical experiments.

**Questions:**

1. I was wondering whether you could provide more intuition to use the Q-value.

2. If the probability of applying the treatment is not 1/2, which might be a small value, how will this change affect those results.

3. I wonder if there are some finite sample results corresponding to Theorem 2?

4. For the comparison between the naive estimator and the DQ estimator, if the difference of the transition probability distributions between applying treatment or not is large, will the naive estimator work better or not?

**Limitations:**

I think there is no potential negative social impact of the work.

**Strengths And Weaknesses:**

Strengths:

1. The idea of this new estimator is simple and clear, and, at the same time, the authors provide strong results.

2. This new estimator is computationally affordable.

Weaknesses:

1. It would be better to have some finite sample results corresponding to Theorem 2.

2. The motivation or intuition of using Q-value instead of reward seems not strong enough.

---

> ### Author Response · Authors · 2022-08-02
> **A response to Reviewer frjR**
>
> ### **Providing more intuition about using $Q$ value**:
> This is an excellent question! In short, our design of the DQ estimator is based on a novel "Taylor"-like expansion for $\rm{ATE}$ as a function of $\delta$. Specifically, writing ${\rm ATE} := f(\delta)$, we observe that a Taylor expansion of $f(\delta)$ can be obtained by iterating a particular perturbation identity for stationary distributions of Markov chains. From here, we discover that:
> 1. Using the "zeroth order" form of this expansion to estimate the values of the intervention yields the Naive estimator (and $O(\delta)$ bias which corresponds to a bound on the first order remainder)
> 2. Using the "first order" form of this expansion yields the DQ estimator (and $O(\delta^2)$ bias which corresponds to a bound on the second order remainder). Now to your question – It turns out that this first order expansion maps precisely to a difference of $Q$-functions.
>
> We have provided a detailed sketch of this approach in our response to Reviewer P5xj, and additional details can be found in Appendix C of the supplementary materials.
>
> ### **If treatment probability is not $\frac{1}{2}$:**
> Suppose the treatment probability for policy 0 is $q$. The estimator presented carries over essentially unchanged, with the following transformation: we compute the Q-function with ``propensity-score-adjusted’’ rewards $r'$: where $r'(s, 0) = r(s, 0)\cdot (1-q)/q, r'(s, 1) = r(s, 1)\cdot q/(1-q)$. Note that since $q$ is an algorithmic choice, the 'propensity-score' here is constant and known so that the transformation is trivial.
> Our results on bias are unchanged, but as one might expect, variance increases by roughly a factor of $O(1/q)$.
>
> ### **Finite-sample results:**
> A natural approach to doing this would be to consider using Berry-Esseen type results to refine the CLT analysis at the heart of our variance characterization. The non-triviality here is (a) characterizing the higher order coefficients in the Berry-Esseen approximation, and (b) justifying that our chain is amenable to the approximation (i.e. these would be conditions on the MDP itself). An alternative to this approach is attempting to extend the exciting recent progress on finite-time analyses of TD (Zhang, Zhang, Maguluri ICML2021, Qiu, Yang, Ye, Wang JSAIT2021). It is unclear that this approach would yield a useful characterization on the constants in the rates obtained however, and these constants are actually essential since trivial characterizations that scale with the size of the state space are of limited value.
>
> ### **When $\delta$ is large**:
> As long as $\delta < 1- \lambda$, where $\lambda$ is the mixing time, the Naive estimator will always have worst case bias that is inferior to the DQ estimator – this can be seen by examining the ‘remainder’ terms in the Taylor-expansion we alluded to earlier.
> In our experiments, we find that the bias of the DQ estimator remains substantially lower than the Naive estimator even as $\delta$ grows large. Specifically, in response to your question, we modified the experiments in Section 5.1 to have $\delta = 0.99$, and the bias of DQ was still 1% that of the Naive estimator.

---

> > ### Comment · Reviewer_frjR · 2022-08-07
> > **Rebuttal Acknowledgement**
> >
> > Thank you very much for your reply! My questions are addressed!

---

### Official Review · Reviewer_7PnB · 2022-07-11

**Rating:** 7
**Confidence:** 2
**Soundness:** 3 good
**Presentation:** 3 good
**Contribution:** 3 good

**Summary:**

The paper considers the problem of estimating the average treatment effect, where experimenting on some treatment group might also affect the control group. The problem is motivated by several practical scenarios, such as inventory management, ride sharing, etc, where both the treatment and the control group share a common pool of resources. The authors cast this as an off-policy evaluation problem and propose a new estimator that adequately balances the bias-variance trade-off in this setting. Experiments on a toy domain and raid-hailing simulator are used to demonstrate the effectiveness of the proposed method.



**Questions:**

A\. Section 2.1: My understanding is that during implementation the intervention policy is at least tried to be coherent within an episode (here episode might correspond to one user or one session). In the discussion/example it seems like the policy is being switched within a single episode. I can see one argument that from the point of view of the system, it may not be able to distinguish different users, but in that case it is a POMDP setting and not an MDP one?

I see the high-level problem but I am quite confused by the formulation of it. (For disclosure, I am not well aware of the interference literature.)

B\. Assumption 1: For which values of $k$? I am not sure if I have a complete understanding of what this assumption implies and how practical it is. Particularly, along with the assumption in Theorem 1, what does it say about the steady state distribution and the mixing time for the treatment and the control policies?

C\. Line 176 and Eqn 3: At both the places the proposed estimator is dropped in without much discussion about why one would expect this to work. I see the proof for Theorem 1, but I am failing to understand intuitively why the proposed procedure helps. Both the Q functions and the distribution for computing the expectations are for the data collecting policy. It feels like this estimator is only making one step correction according to the two policies being compared, which is a lot like what the naive estimator would do as well. What is the correct way to interpret this estimator?


D\. How does the mixing time assumption influence the empirical results? Showing this on the toy-example at least would be helpful.


**Limitations:**

While the proposed method is straightforward to express, I do not think I have a good understanding of it to provide any constructive criticism (I had not bid to review this work). As such, I have lowered my confidence for this review and will update my score based on discussions with other reviewers after the rebuttal.

**Strengths And Weaknesses:**

Strengths

- The problem is applicable to a large number of practical applications and thus the work can be of high relevance to a broad audience.
- The proposed estimator is straightforward to implement, and at the same time provides adequate bias-variance trade-off.
- The proposed method provides strong performance in the experiments conducted.

Weakness

- It is not clear how reasonable is Assumption 1.

---

> ### Author Response · Authors · 2022-08-02
> **A Response to Reviewer 7PnB**
>
> Thank you for the helpful review! In response to your questions:
>
> ### **Coherent Policy Within an Episode**
> To clarify the formulation, and the issue of interference: In a typical problem setting, a user arrives at each time step. Then, with probability $p$, the user receives the treatment (i.e. the system takes the action prescribed by the intervention policy), and with probability $1-p$, the user receives the control (i.e. the system takes the action prescribed by the incumbent policy). Following this, the state of the system evolves (based potentially on the users response), thereby impacting subsequent arrivals to the system. The system is expected to mix, but there is no concept of a ‘reset’ or ‘episode’. This is the setting described in the introduction.
>
> If the actual identity of a user is relevant to payoffs, then the state of the chain can be expanded to capture said identities (or relevant user features). Persisting treatments for the same user arriving at multiple timesteps can also be handled in this way, by incorporating identity into the state description.
>
> ### **Assumption 1**
> Assumption 1 requires the inequality hold for all $k$ and some $\lambda < 1$ -- sorry for not being more precise. In fact, this is benign: a general, sufficient condition for Assumption 1 to hold is that the Markov chain is ergodic (i.e., irreducible and aperiodic). We can clarify this in the manuscript. This is a **very** general class of Markov chains: in practical terms, this models essentially any Markovian system as long as no action leads to an “irrecoverable” state (i.e., all states are always reachable in finite time). Note that most of the average-reward MDP literature makes this assumption (see e.g. this recent lit review https://arxiv.org/pdf/2010.08920.pdf). Finally, Theorem 1 holds for any $\delta$ and requires Assumption 1 hold; nothing is required of the mixing times of steady-state distributions of the treatment or control policies.
>
> ### **Intuition for the DQ estimator**
> This is an excellent question! In short, our design of the DQ estimator is based on a novel `Taylor’-like expansion for $\rm{ATE}$ as a function of $\delta$. Specifically, writing ${\rm ATE} := f(\delta)$, we observe that a Taylor expansion of $f(\delta)$ can be obtained by iterating a particular perturbation identity for stationary distributions of Markov chains. From here, we discover that:
>
> 1. Using the "zeroth order" form of this expansion to estimate the values of the intervention yields the Naive estimator (and $O(\delta)$ bias which corresponds to a bound on the first order remainder)
> 2. Using the "first order" form of this expansion yields the DQ estimator (and $O(\delta^2)$ bias which corresponds to a bound on the second order remainder). Now to your question – It turns out that this first order expansion maps precisely to a difference of $Q$-functions.
>
> We have provided a detailed sketch of this approach in our response to Reviewer P5xj, and additional details can be found in Appendix C of the supplementary materials.
>
> ### **Mixing Time and Practical Performance**
> This is a great question. In the ride-sharing example, the quantity $1/(1-\lambda)$ translates into about $10^5$ time steps (i.e. $\lambda$ is very close to 1). Despite this, we see that the bias of the DQ estimator remains low, and in particular much lower than the Naive estimator. We can discuss this further in the update.

---

> > ### Comment · Reviewer_7PnB · 2022-08-09
> > **Re:**
> >
> > Thank you for your clarification. Adding this discussion about the assumption and more intuition for the proposed method in the main paper can be beneficial. I have no other questions, and I have updated the score accordingly.

---

### Official Review · Reviewer_P5xj · 2022-07-12

**Rating:** 8
**Confidence:** 4
**Soundness:** 3 good
**Presentation:** 4 excellent
**Contribution:** 4 excellent

**Summary:**

This paper examines estimating the difference in reward performance of two policies for a Markov Decision Process (MDP) by running a mixture policy and performing on-policy evaluation. The paper shows theoretically the benefits over current methods of policy evaluation of using a difference in $Q$ values estimation process. Specifically, this approach obtains a small estimator variance and a estimation bias that is second order in the effect of the "intervention" policy (the policy being tested against an existing policy).

**Questions:**

Line 38: "designs are not be an ideal lever";  perhaps "be" should be deleted.

Line 38 and 39: "and often infeasible"; perhaps this should be "and are often infeasible" .



**Limitations:**

It would have been nice to see some proof outlines in the paper. There was no supplementary appendix at the end of the paper. I assume it required a separate download.

**Strengths And Weaknesses:**

(S) the paper is very well written. It is clear and precise. I found very few typos.

(S) The paper tackles a very  important problem and gives a compelling argument and demonstration for how it can be solved in a much better fashion compared to current approaches.

(S) Theoretical results are explained well (NOTE: although no proofs are included in the body of the paper).

(S) The experimental results use a large scale realistic problem and shown strong evidence for substantial and meaningful improvement.

(S) The experimental results are very well presented and interpreted.

(S) Excellent reference list (66 references).

(W) It would have been nice to see proof outlines (at least) in the paper. I assume proofs are given in supplemental material ( I did not check).

---

> ### Author Response · Authors · 2022-08-02
> **Intuition for the DQ estimator and a proof outline**
>
> Thank you for the extremely positive and helpful review! We plan to add proof outlines to the main body.
>
> ### **Intuition for the DQ estimator**
> This is an excellent question! In short, our design of the DQ estimator is based on a novel `Taylor’-like expansion for $\rm{ATE}$ as a function of $\delta$. Specifically, writing ${\rm ATE} := f(\delta)$, we observe that a Taylor expansion of $f(\delta)$ can be obtained by iterating a particular perturbation identity for stationary distributions of Markov chains. From here, we discover that:
> 1. Using the "zeroth order" form of this expansion to estimate the values of the intervention yields the Naive estimator (and $O(\delta)$ bias which corresponds to a bound on the first order remainder)
> 2. Using the "first order" form of this expansion yields the DQ estimator (and $O(\delta^2)$ bias which corresponds to a bound on the second order remainder). Now to your question – It turns out that this first order expansion maps precisely to a difference of $Q$-functions.
>
>
> ### **Proof Outline of Theorem 1**
> (Preliminary notation: let $A^{\\#}$ denote the *group inverse* of a matrix $A$, which is one type of inverse that is useful in the MDP context.) To begin, let us state two facts:
>
> **Fact 1.** For any policy $\pi$ and associated transition matrix $P_{\pi}$ and state-wise rewards $r_{\pi}$, the bias function $V_{\pi}$ of $\pi$ has the following **explicit** formula:
> $$
> V_{\pi} = (I-P_{\pi})^{\\#} r_{\pi}.
> $$
>
> **Fact 2.** For any policies $\pi, \pi'$ with corresponding stationary distributions $\rho_{\pi}, \rho_{\pi'}$ and transition matrices $P_{\pi}$ and $P_{\pi'}$, we have the following perturbation **identity** for stationary distributions (Meyer 1980):
> $$
> \rho_{\pi}^{\top} = \rho_{\pi'}^{\top} + \rho_{\pi}^{\top}(P_{\pi}-P_{\pi'})(I-P_{\pi'})^{\\#}.
> $$
>
> We use these two facts to obtain a Taylor expansion for $\rm{ATE}$. Note that $\rm{ATE} = \rho_{1}^{\top}r_1 - \rho_0^{\top}r_0$, where $\rho_{1} \in R^{|S|}, \rho_{0} \in R^{|S|}$ are the stationary distributions of $P_1$ and $P_0$ respectively; and $r_{1}, r_{0}$ are the reward vectors associated with the actions $1$ and $0$ respectively. We further set $P_{1} - P_{0} = \delta A$ where $\delta \in R$ is chosen such that the row absolute sum of $A$ is bounded by 1.
>
> We analyze $\rho_{1}^{\top}r_1$ first; $\rho_0^{\top}r_{0}$ follows by symmetry. Applying Fact 2 to $\rho_{1}$ based on $\rho_{1/2}$, we have
> \begin{align*}
> \rho_{1}^{\top}r_1 &= \rho_{1/2}^{\top}r_1 + \rho_{1}^{\top}(P_{1}-P_{1/2})(I-P_{1/2})^{\\#}r_{1} \\\\
> &=  \rho_{1/2}^{\top}r_1 + \frac{\delta}{2}\rho_{1}^{\top}A(I-P_{1/2})^{\\#}r_{1}.
> \end{align*}
> Applying Fact 2 to expand $\rho_{1}$ again using $\rho_{1/2}$ in the RHS of the above equation:
> \begin{align*}
> \rho_{1}^{\top}r_1
> &=  \rho_{1/2}^{\top}r_1 + \frac{\delta}{2}\rho_{1}^{\top}A(I-P_{1/2})^{\\#}r_{1}\\\\
> &=  \rho_{1/2}^{\top}r_1 + \frac{\delta}{2}\rho_{1/2}^{\top}A(I-P_{1/2})^{\\#}r_{1} +  \frac{\delta^2}{4}\rho_{1}^{\top}\left(A(I-P_{1/2})^{\\#}\right)^2 r_{1}
> \end{align*}
> This can be iterated so that the $k$-th order expansion, in terms of O($\delta^{k}$), can be obtained. But the second order is sufficient for the analysis here. A similar analysis can be applied for $\rho_{0}^{\top}r_0$ and after cleaning this up, we obtain
> \begin{align*}
> \rm{ATE}
> &= \rho_{1}^{\top}r_1  - \rho_{0}^{\top}r_0\\\\
> &=  \rho_{1/2}^{\top}(r_1-r_0) + \delta \cdot \rho_{1/2}^{\top} A (I-P_{1/2})^{\\#} \frac{r_1+r_0}{2} + O(\delta^2).
> \end{align*}
>
> **Naive Estimator: zeroth-order expansion**: From the above, it is clear that the zeroth order term above $\rho_{1/2}^{\top}(r_1-r_0)$ exactly corresponds to the Naive estimator, which naturally has $\Omega(\delta)$ bias in general.
>
> **DQ estimator: first-order expansion**: To see why the first order approximation corresponds to our DQ estimator, we can use Fact 1, i.e., $(I-P_{1/2})^{\\#} r_{1/2} = V_{1/2}$, and recall that $\delta A = P_1 - P_0$, then
> \begin{align}
> &\rho_{1/2}^{\top}(r_1-r_0) +  \delta \cdot \rho_{1/2}^{\top}  A (I-P_{1/2})^{\\#} \frac{r_1+r_0}{2}  \\\\
> &= \rho_{1/2}^{\top}(r_1-r_0) + \rho_{1/2}^{\top} (P_1-P_0) V_{1/2} \\\\
> &= \rho_{1/2}^{\top}(r_1 - r_0 + (P_1-P_0)V_{1/2})\\\\
> &=\rho_{1/2}^{\top} (Q_1 - Q_0),
> \end{align}
> where the last equality uses the definition of the $Q$-function. The formula is somewhat surprisingly elegant. As an aside, this also suggests how to construct arbitrary kth-order expansions for further bias reduction, although potentially at the cost of increased variance. See Appendix C in the supplementary materials for details.

---

> > ### Comment · Reviewer_P5xj · 2022-08-08
> > **Thank you for the additional information.**
> >
> > Adding insights like that given in your proof outline will further strengthen the paper.

---

### Official Review · Reviewer_RgZV · 2022-07-15

**Rating:** 7
**Confidence:** 3
**Soundness:** 4 excellent
**Presentation:** 4 excellent
**Contribution:** 3 good

**Summary:**

The paper introduces a treatment estimator for experiments with markovian interference using an on-policy RL approach. The authors show that their estimator achieves a good bias variance tradeoff vs other baselines (two sided randomization, off policy estimators), and that unbiased estimators necessarily have large variance. Finally, the technical results and estimation strategies are illustrated in two experiments: on that re-uses the framework from Johari et al, and another the author's own simulation code that simulates a ride-sharing experiment.

**Questions:**

The averaging of estimates over long periods of time into single point estimates is not always regular practice, and I'm afraid that in this case, the framework wouldn't apply so easily. Often, just daily estimates are compiled and reported. Is this because of non markovian interference, or seasonal effects, or time to reach equilibrium? Some discussion there would clarify things for the reader and map this to a setting they are familiar with.

**Strengths And Weaknesses:**

The paper seemed to me to be strong. It's well written and was enjoyable to read. I have little criticism to make, though I have to admit I'm not familiar enough with the RL literature to be entirely confident in my assessment. I particularly appreciated that the experiment section was clear, and re-used available open source code, or made the code available, with sufficient baselines.

 One thing that would improve the paper is to relate the notation to the potential outcomes notation which is standard in this setting and which many readers in this space are familiar with. More specifically, it would be nice to characterize in this notation which settings can be considered markovian interference where this framework applies.

nit: typos l. 38, l. 68.

---

> ### Author Response · Authors · 2022-08-02
> **A Response to Reviewer RgZV**
>
> Thank you for your review! Please find a response to your two questions below; we can certainly include some of this discussion in the paper.
>
> ### **Potential Outcomes**
> There are two approaches we could take here to cast the setup within Potential Outcomes. The first is similar to the approach taken in Bajari et. al. 2020, where the domain of the potential outcome function is high dimensional (in that case, depending on the treatment of a subset of other units). Specifically, in our setting, the domain of the potential outcome function would be the sequence of actions taken. However, this is not an ideal approach (as observed already in Bajari et. al.) since it does not yield an approach to approximating ATE with experimental outcomes.
>
> An alternate approach is to posit potential outcomes that are *state-dependent*. This is akin to a more traditional view of the potential outcomes framework with the exception that the distribution over states as opposed to being exogenous (a situation which would trivially lend itself to potential outcomes), is now endogenous. Our work can be viewed as characterizing the precise bias introduced by this endogeneity.
>
> ### **Daily estimates**
>
> One approach to addressing temporal interference in general is indeed a switchback design, where, say, treatment and control are applied on alternate days, and then the difference in the "daily estimates" constitutes a treatment effect estimate. There are two challenges to leveraging this design. The first, is that this design requires that for some period of time, the entire system be subject to the intervention, which in some applications is not viewed as an acceptable risk. The second, more significant challenge, as you rightly recognize, is that the system will often not mix fast enough (i.e. `get to equilibnrium’ as you put it). Our work can be viewed as a solution to this challenge, when it is possible to identify a notion of ‘state’ with respect to which the underlying system is Markovian.

---

### Meta-Review · Area_Chair_qPZp · 2022-08-26

**Recommendation:** Accept
**Confidence:** Less certain

**Metareview:**

Please add proof outlines to the main body in the final version. Also add a discussion on Assumption 1 and more insights for the proposed method.

**Award:**

No

---

### Decision · Program_Chairs · 2022-09-14

Accept